# Signal Domain Learning Approach for Optoacoustic Image Reconstruction from Limited View Data

**Anna Klimovskaia Susmelj**[*1]                                   ANNA@KLIMOVSKY.RU
**Berkan Lafci**[*2,3]                                              BLAFCI@ETHZ.CH
**Firat Ozdemir**[1]                                               FIRAT.OZDEMIR@SDSC.ETHZ.CH
**Neda Davoudi**[2,3]                                              DAVOUDIN@STUDENT.ETHZ.CH
**Xosé Luís Deán-Ben**[2,3]                                        XL.DEANBEN@PHARMA.UZH.CH
**Fernando Perez-Cruz**[1,4]                                       FERNANDO.PEREZCRUZ@SDSC.ETHZ.CH
**Daniel Razansky**[†2,3]                                          DANIEL.RAZANSKY@UZH.CH

[1] *Swiss Data Science Center, ETH Zürich and EPFL, Zürich, Switzerland*

[2] *Institute of Pharmacology and Toxicology and Institute for Biomedical Engineering, Faculty of Medicine, University of Zurich, Switzerland*

[3] *Institute for Biomedical Engineering, Department of Information Technology and Electrical Engineering, ETH Zurich, Switzerland*

[4] *Institute for Machine Learning, Department of Computer Science, ETH Zurich, Switzerland*

**Editors:** Under Review for MIDL 2022

## Abstract

Multi-spectral optoacoustic tomography (MSOT) relies on optical excitation of tissues with subsequent detection of the generated ultrasound waves. Optimal image quality in MSOT is achieved by detection of signals from a broad tomographic view. However, due to physical constraints and other cost-related considerations, most imaging systems are implemented with probes having limited tomographic coverage around the imaged object, such as linear array transducers often employed for clinical ultrasound (US) imaging. MSOT image reconstruction from limited-view data results in arc-shaped image artifacts and disrupted shape of the vascular structures. Deep learning methods have previously been used to recover MSOT images from incomplete tomographic data, albeit poor performance was attained when training with data from simulations or other imaging modalities. We propose a two-step method consisting of i) style transfer for domain adaptation between simulated and experimental MSOT signals, and ii) supervised training on simulated data to recover missing tomographic signals in realistic clinical data. The method is shown capable of correcting images reconstructed from sub-optimal probe geometries using only signal domain data without the need for training with ground truth (GT) full-view images.

**Keywords:** Optoacoustics, Limited View Artifacts, Signal Domain Learning, Style Transfer, Domain Adaptation.

## 1. Introduction

Multi-spectral optoacoustic tomography (MSOT) is a hybrid biomedical imaging modality based on optical excitation (thermal expansion) of biological tissues followed by detection of the generated ultrasound (US) waves. Interpretation and quantification of the MSOT data

---

[*] Contributed equally

[†] Correspondence

is often hampered by poor tomographic coverage provided by the common clinical imaging system implementations. Optimal tomographic inversion in MSOT implies recording sufficient information from the generated ultrasonic wave field with broad angular tomographic coverage (Merčep et al., 2015). Yet, full tomographic coverage is generally not possible due to physical constraints and other cost-related considerations.

The potential value of MSOT as a clinical imaging tool can be further enhanced when combined with pulse-echo (reflection) US imaging, which provides important reference anatomical information (Merčep et al., 2019; Lafci et al., 2020). However, such combination is not straightforward cause MSOT and US imaging impose different, often contradictory, constraints on the transducer array design, such as the pitch size or element directivity in transmission and reception. Linear arrays are frequently used in commercial hand-held US scanners and have further been suggested for implementation of hybrid optoacoustic-ultrasound (OPUS) scanners (Jeng et al., 2021; Montilla et al., 2012). Linear arrays are easy to manufacture and well-established guidelines exist for the interpretation of the generated US images, offering clear advantages in the clinical setting. On the other hand, acquisition of MSOT images with linear arrays typically results in elongated vessel structures and arc-shaped limited-view artifacts, thus making image interpretation and quantification difficult (Deán-Ben and Razansky, 2016). Recently, specialized transducer geometries have been suggested for optimal implementation of hybrid OPUS scanners, such as multisegment transducer arrays incorporating both linear and concave array segments. In this way, the limited-view MSOT problem can be partially mitigated with image quality restored to a certain degree (Fig. 1A) (Deán-Ben et al., 2017). In the multisegment configuration, the linear part renders standard US images while all elements contribute to an increased angular coverage for MSOT image reconstruction (Merčep et al., 2017). While providing optimal image quality in both MSOT and US modes, the manufacturing process of an array of this type is relatively complex. Furthermore, hand-held scanning of certain parts of the human body is also hampered with this array geometry due to the need for a customized water coupling approach (Merčep et al., 2018).

A possible approach to resolve these unavoidable trade-offs is to make use of the optimized multisegment array configuration for the development of learning-based methods toward improving image quality in both MSOT and US imaging modes. In this work, we approach the hybrid image reconstruction problem by considering the raw signal domain data instead of dealing directly with the reconstructed images. With this, we aim to capture spatial and temporal correlations between the transducer array elements in latent space. A learning-based method trained on simulated data may be used to complete the missing view angles in the linear array recordings. To this end, deep learning methods have been used to partially restore quality of the MSOT images reconstructed from incomplete tomographic data. However, poor performance was achieved when training with data from simulations or other imaging modalities, which appears to be a result of the large domain gap between simulated and experimental data (Davoudi et al., 2019). Thereby, we propose a two-step approach consisting of i) style transfer for domain adaptation between simulated and experimental MSOT signals, and ii) semi-supervised training on simulations from multisegment array geometry and experimental data from linear array to recover missing signals in experimental multisegment array data. We formulate the domain adaptation problem as an unpaired image translation between simulated and experimental signals. Signals detected

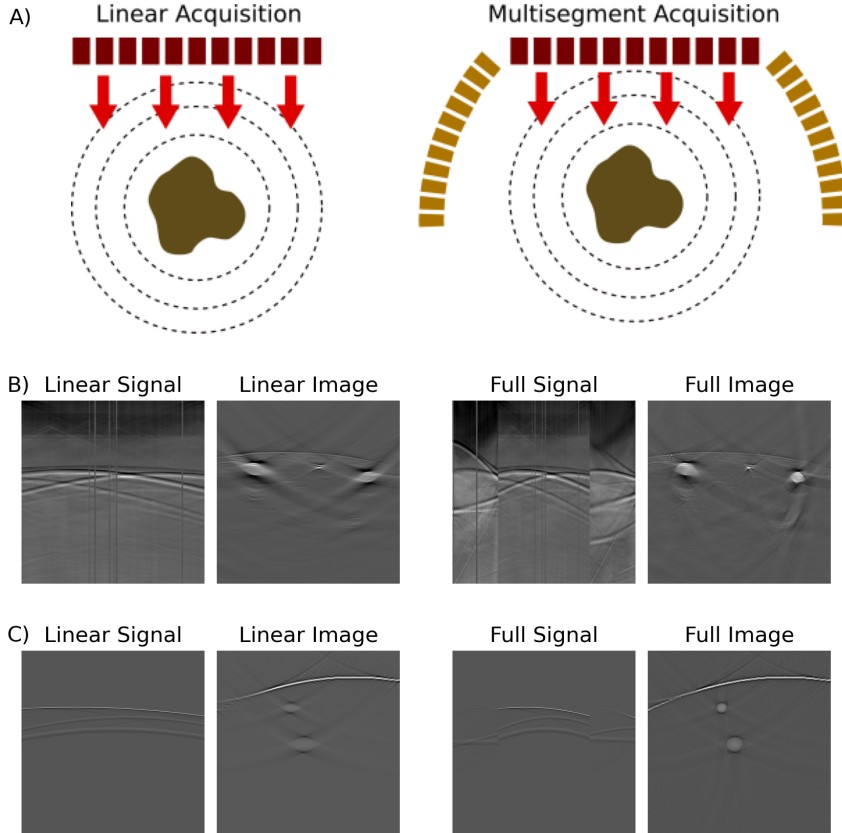

Figure 1: Handheld MSOT imaging with linear and multisegment array configurations. A) Schematic diagram of the array geometries. The excitation light beam and generated US waves are represented with red arrows and dashed circles, respectively. B) Raw data (time-resolved signals) along with the reconstructed MSOT images corresponding to a hand-held scan of the human arm at 1064 nm excitation wavelength with the linear and multisegment arrays, respectively. C) Simulated signals along with the reconstructed images for the linear and multisegment arrays, respectively.

by elements of the concave segments are then estimated from the data provided by the linear part of the array using only simulated data after domain adaptation. Once the missing parts have been recovered, the MSOT image reconstruction can be performed with standard methods such as back-projection or model-based algorithms (Ozbek et al., 2013; Xu and Wang, 2005; Ding et al., 2020). The main contributions of this paper with respect to prior art are summarized as follows: i) the MSOT limited-view problem is tackled in the signal domain instead of the image domain, ii) a method to reduce domain gap between simulated and experimental data is proposed, and iii) a learning-based method is used for estimating the signals from missing detection elements without the need for ground truth (GT) data from sophisticated and costly array configurations (e.g. multisegment array). To the best of our knowledge, this work constitutes the first attempt to address the limited-view MSOT problem in the raw signal domain using simulated data.

## 2. Methodology

### 2.1. Style Transfer Network

The first step of the proposed method represents the key component to reduce the domain gap between simulated and experimental data. The main differences between simulated and experimental domains result from difficulties in mimicking complicated acoustic pressure wavefields corresponding to actual anatomical structures and from the lack of realistic noise components in simulated data. Herein, we use a style transfer network to reduce the distance between simulated and experimental datasets. The simulated and experimental domains are denoted as X and Y, respectively. The corresponding training samples are denoted as $\{x_i\}_{i=1}^N$, $\{y_i\}_{i=1}^M$ where N and M are the number of samples from simulated and experimental data, respectively. The signals from both domains are combined into one dataset alongside their labels, so that each signal is represented by a pair $\mathcal{D} = \{(s_i, l_i)\}_{i=1}^{N+M}$, where $s_i$ is the signal itself and $l_i$ is a label indicating if the signal belongs to the simulated ($l_i = 0$) or experimental ($l_i = 1$) domains.

An encoder-decoder architecture was used with domain-adversarial training on the latent space from fader networks (Lample et al., 2017) after minor modifications. As demonstrated in Fig. 2A, the model consists of two convolutional neural networks; an encoder $E_{\theta_{\text{enc}}}$ and a decoder $D_{\theta_{\text{dec}}}$, a latent space discriminator and an additional fully connected discriminator network $D_{\theta_{\text{disc}}}$ for the adversarial training to ensure domain alignment. The encoder $E_{\theta_{\text{enc}}}$ takes as input the signal 2D representation $s_i$ and produces the latent representation $z_i = E_{\theta_{\text{enc}}}(s_i)$. The decoder $D_{\theta_{\text{dec}}}$ takes as input the latent invariant representation $z_i$ and corresponding label of the input domain $l_i$ to produce the reconstructed signal $\hat{s}_i$. We use mean absolute error (MAE) as the reconstruction error as it is more suitable for medical imaging problems since it produces sharper images:

$$\ell_{\text{MAE}} = \frac{1}{N + M} \sum_{(s,l) \in \mathcal{D}} \|D_{\theta_{\text{dec}}}(E_{\theta_{\text{enc}}}(s), l) - s\|_1. \tag{1}$$

The decoder takes as an input latent representation and a label. If the label is "experimental", a latent vector is sampled from a Gaussian distribution with mean and variance parametrized by embedding layers in the decoder, and then a convolutional network is applied to it. If the label is "simulated", another convolutional network is applied directly to the latent representation without sampling. The latent discriminator $D_{\theta_{\text{disc}}}$ is trained to make this representation $z_i$ invariant to the domain via an adversarial loss as in GANs (Goodfellow et al., 2014). In particular, this is achieved by a two-player game; the discriminator is trained between two domains with a classification loss:

$$\ell_{\text{latent\_disc}} = -\mathbb{E}_{x \sim P_{\text{exp}}}[\log D_{\theta_{\text{disc}}}(E_{\theta_{\text{enc}}}(x))] - \mathbb{E}_{y \sim P_{\text{sim}}}[\log(1 - D_{\theta_{\text{disc}}}(E_{\theta_{\text{enc}}}(y)))], \tag{2}$$

and the parameters of the encoder are optimized through an additional adversarial loss:

$$\ell_{\text{adv\_latent}} = -\mathbb{E}_{s \sim P_s}[\log D_{\theta_{\text{disc}}}(E_{\theta_{\text{enc}}}(s))], \tag{3}$$

where $P_{\text{exp}}$ and $P_{\text{sim}}$ are the experimental and simulated data distributions, and $P_s$ is the joint distribution of both experimental and simulated data.

Since adversarial training can be unstable (Arjovsky and Bottou, 2017; Bao et al., 2017;

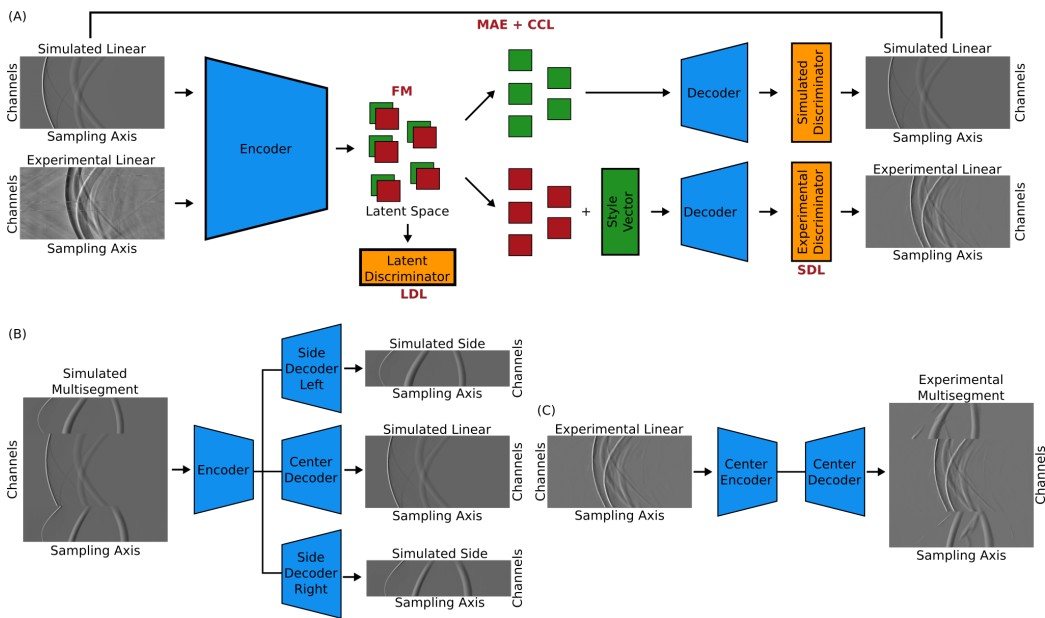

Figure 2: Summary of the proposed network architectures. A) Style network architecture with loss functions defined by red labels. MAE - Mean Absolute Error, CCL – Cycle Consistency Loss, FM – Feature matching loss, LDL – Latent Discriminator Loss, SDL – Style Discriminator Loss. B) Training of side network with simulated data. C) Training of side network using only experimental data from the linear part of the array.

Arjovsky et al., 2017), we additionally add feature matching to stabilize adversarial training(Bao et al., 2017):

$$\ell_{\text{FM}} = \frac{1}{2}\|\mathbb{E}_{x \sim P_{\text{exp}}}[E_{\theta_{\text{enc}}}(x)] - \mathbb{E}_{y \sim P_{\text{sim}}}[E_{\theta_{\text{enc}}}(y)]\|_2^2. \tag{4}$$

We opt for patchGAN discriminator (Zhu et al., 2017) as proposed by Lample et al. (2017) to improve the quality and sharpness of the reconstructions where the discriminators are trained by updating following two loss functions:

$$\ell_{\text{sim}} = -\mathbb{E}_{s \sim P_{\text{exp}}}[\log D_{\theta_{\text{disc\_sim}}}(D_{\theta_{\text{dec}}}(E_{\theta_{\text{enc}}}(s), l = \text{sim}))], \tag{5}$$

$$\ell_{\text{exp}} = -\mathbb{E}_{s \sim P_{\text{sim}}}[\log D_{\theta_{\text{disc\_exp}}}(D_{\theta_{\text{dec}}}(E_{\theta_{\text{enc}}}(s), l = \text{exp}))]. \tag{6}$$

Patch discriminators $D_{\theta_{\text{disc\_sim}}}$ and $D_{\theta_{\text{disc\_exp}}}$ are trained by minimizing the following losses respectively:

$$\ell_{\text{sim\_disc}} = -\mathbb{E}_{x \sim P_{\text{sim}}}[\log D_{\theta_{\text{disc\_sim}}}(x)] - \mathbb{E}_{y \sim P_{\text{exp}}}[\log(1 - D_{\theta_{\text{disc\_sim}}}(D_{\theta_{\text{dec}}}(E_{\theta_{\text{enc}}}(y), l = \text{sim})))], \tag{7}$$

$$\ell_{\text{exp\_disc}} = -\mathbb{E}_{x \sim P_{\text{exp}}}[\log D_{\theta_{\text{disc\_exp}}}(x)] - \mathbb{E}_{y \sim P_{\text{sim}}}[\log(1 - D_{\theta_{\text{disc\_exp}}}(D_{\theta_{\text{dec}}}(E_{\theta_{\text{enc}}}(y), l = \text{exp})))]. \tag{8}$$

The overall training procedure consists in iteratively updating the generator networks via minimization of the total loss

$$\ell_{\text{total}} = \ell_{\text{MAE}} + \ell_{\text{adv\_latent}} + \ell_{\text{Cycle}} + \ell_{\text{FM}} + \ell_{\text{sim}} + \ell_{\text{exp}}, \tag{9}$$

where $\ell_{\text{Cycle}}$ is cycle consistency loss (Isola et al., 2017) and the discriminators are updated every $n$ epochs via minimizing losses from equations 3, 7, 8 separately. This combination of losses was originally proposed in Lample et al. (2017), except the cycle loss, which was proposed for domain adaptation in Hoffman et al. (2018). Such a particular combination of losses is important for a good quality of predictions for domain adaptation task. We will refer to this network as style network in the rest of this manuscript. The fully trained style network will be denoted as $S_{\theta_{Style}}$ with $s_{i,lin}$ linear parts of a signal $i$ and $s_{i,mul}$ multisegment array detection of this signal.

## 2.2. Side Network

After reducing the domain gap between simulated and experimental data, a second (side) network of auto-encoders is suggested to overcome limited-view-associated problems and yield geometrically corrected images. The main goal of this side network is to impute virtual signals at the concave parts of the multisegment array (Fig. 1A) using side decoders. Specifically, one encoder and three decoders are used in the auto-encoder architecture for simulated signals Fig. 2B. It is important to note that only the central part of the network (encoder and center decoder) is used in the training phase when the observation corresponds to an experimental signal. By training the network on simulated data, this is optimized to complete the concave sides of the imaging array. Accordingly, experimental signals are included in the dataset for optimization of the encoder and center decoder. Training on linear experimental signals helps to adapt network for experimental data distribution.

In a similar manner as in the style network, we propose an encoder-decoder architecture. As in $S_{\theta_{Style}}$, the encoder takes the signal matrix corresponding to linear array from either simulated or experimental images. The decoder, in contrast to $S_{\theta_{Style}}$, consists of three convolutional networks, where the goals of each networks are to produce signals from i) linear array ("center decoder"), ii) left concave segment, and iii) right concave segment. During training, the network only sees linear parts of experimental signals and whole multisegment array signals for simulated data. We use MAE loss to train the networks:

$$\ell_{sides} = \frac{1}{N} \sum_{(s_{lin}, s_{mul}) \in \mathcal{D}_{sim}} \|D_{\theta_{dec}}(E_{\theta_{enc}}(S_{\theta_{Style}}(s_{lin}))) - s_{mul}\|_1 +$$

$$\frac{1}{M} \sum_{s_{lin} \in \mathcal{D}_{exp}} \|D_{\theta_{dec}}(E_{\theta_{enc}}(S_{\theta_{Style}}(s_{lin}))) - s_{lin}\|_1. \quad (10)$$

## 3. Experiment and Results

### 3.1. Datasets description

The datasets used in this study include two main parts. The simulated dataset contains a curved structure mimicking the skin surface and circular shapes similar to round vessel structures in the human forearm. It has 5500 cross-sectional images with different positioning of structures and number of vessels. The simulations were drawn as acoustic pressure maps in the spatial domain. Then, the corresponding signals were generated using the MSOT forward model for the multisegment array geometry (Dean-Ben et al., 2012b). 32

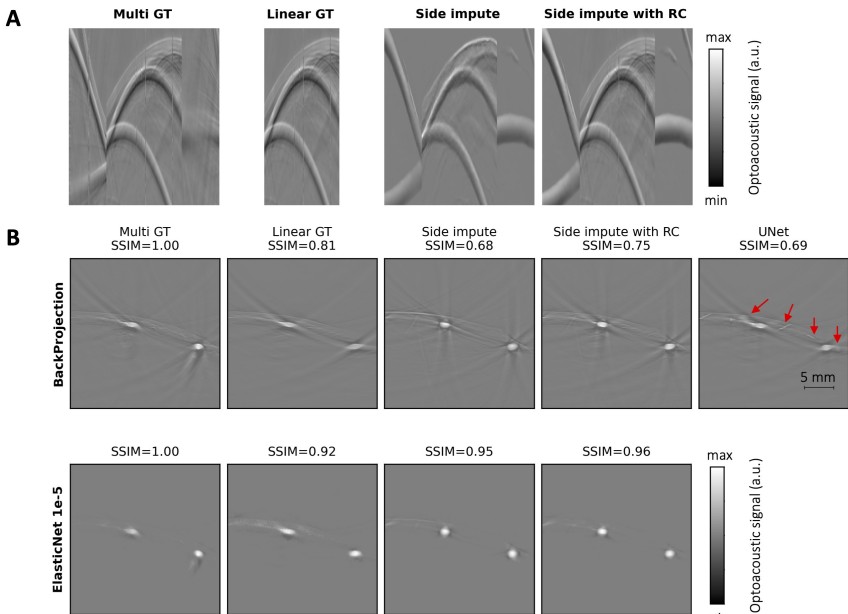

Figure 3: Results in the signal and image domains. A) Signal domain representation of a test image. Left to right: multisegment ground truth (GT) signals, linear GT signals, multisegment signals after style and side networks, multisegment signals after style and side network with real center (RC). B) Images reconstructed with back-projection and model-based elastic-net approach. Benchmark UNet result is added for comparison.

images were hold out for testing. The rest of the images were split into training and validation via 70/30 split. The second dataset was acquired from volunteers using a multisegment ultrasound transducer array as shown in Fig. 1A and described in detail in (Merčep et al., 2017). The corresponding simulated and experimental signals and images from the linear and all parts of the multisegment arrays are shown in Fig. 1B and 1C, respectively. In total, 5565 cross-sectional experimental images were collected from 22 forearms. 5501 images were allocated for training, 32 images for validation and 32 images for testing.

## 3.2. Results

The experimental signals acquired with the linear part were first processed with the style network to generate their virtual simulated counterparts in order to reduce the domain gap with actual simulated data. Then, these simulated signals were fed into the side network to impute the missing concave parts. The signals from the side network and the GT multisegment acquisitions were reconstructed using filtered back-projection and elastic-net algorithms (Ozbek et al., 2013; Dean-Ben et al., 2012b; Zou and Hastie, 2005) to generate the corresponding images. The comparison was made in the image domain because some of the samples from signal domain are not used in reconstruction as they stay out of the FOV or otherwise modified/filtered by pre-processing algorithms before reconstruction. Another reason to evaluate the results in the image domain is that the main goal of this work is to

enhance MSOT images by eliminating limited-view.

One example from test set is shown in Fig. 3. The third column in Fig. 3 shows the output of both networks after style network and side network. We refer to this output as "Sides impute" further in the paper. The linear part of this "Sides impute" output is further replaced with experimental linear part (Fig. 3A - 4th column). We refer to it further in the text as "Sides impute with RC" (RC stands for real center). The corresponding reconstructed images for each signal are displayed in the same column. The network results (side impute and side impute with RC) show clear improvement in round vessel structures which are the most important features visualized in MSOT images. In contrast, benchmark UNet is not able to correct the vessel shapes while further creating additional vessel-like structures.

The quantitative results calculated on the test set are summarized in Table 1 and a complete ablation study in Supplementary Tables 1, 2. Four different metrics were used to evaluate the proposed networks, namely structural similarity index (SSIM), mean squared error (MSE), Pearson correlation coefficient (Pearson) and peak signal to noise ratio (PSNR). The linear GT represents the images reconstructed from cropped multisegment signals; i.e., excluding concave parts. Hence, the linear GT may outperform some of the metrics. However, when the results are compared with the benchmark UNet, the proposed method becomes superior in each evaluation metric.

Table 1: Reconstruction scores with respect to GT multisegment reconstructions for Elastic Net (EN) with $\alpha = 1e - 5$ and BackProjection (BP) methods. Best score is in bold.

|  | SSIM | | MSE | | Pearson | | PSNR | |
| --- | --- | --- | --- | --- | --- | --- | --- | --- |
|  | EN | BP | EN | BP | EN | BP | EN | BP |
| Linear GT | 0.85 | **0.67** | 0.0019 | **0.0013** | 33.75 | **35.37** | 75.86 | **77.48** |
| Ours | **0.90** | 0.64 | **0.0013** | 0.0014 | **35.28** | 35.11 | **77.39** | 77.22 |
| Unet | 0.57 | 0.49 | 0.0026 | 0.0036 | 30.92 | 32.14 | 74.25 | 73.03 |

## 4. Conclusion

This work is the first to pursue a signal domain solution to overcome limited-view reconstruction artifacts in MSOT imaging. Style transfer network was shown to reduce the domain gap between simulations and experimental signals, thus significantly improving quality of the reconstructed images versus conventional learning-free methods, such as back-projection or regularized model-based reconstruction. Previously-suggested methods trained exclusively with simulated data in the image domain (e.g. benchmark UNet) have shown inferior performance as compared to the domain adaptation networks proposed here. It was additionally demonstrated that, once the domain gap is reduced, training with simulated data can be used for imputing missing signals over a broader tomographic angle, thus leading to reduction of limited-view artifacts in backprojection-based reconstructions. However, the proposed method yields slight improvements in backprojection reconstructions. The results can be improved by using different architectures for the similar signal domain approach. Future work will make use of different training invariant architectures or other reconstruction methods.

## Acknowledgments

This work was supported by Swiss Data Science Center (grant C19-04).

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

## Appendix A. Related Work

### A.1. Domain Generalization and Adaptation

Domain adaptation and generalization address the problem of domain shift for settings where the distribution of observations for training and testing differ substantially. A popular approach to address the co-variate shift problem is to minimize the domain gap between two distributions in the latent space by minimizing maximum mean discrepancy (Muandet et al., 2013; Ghifary et al., 2016, 2014; Long et al., 2017b), and adversarial feature alignment (Li et al., 2018; Long et al., 2015; Ganin et al., 2016; Long et al., 2017a; Tzeng et al., 2017). Recently, several methods based on image-level translation for domain adaptation and reduction of style bias have been proposed (Hoffman et al., 2018; Murez et al., 2018; Nam et al., 2021). In particular, Hoffman et al. (2018) proposed using core concepts from CycleGan (Zhu et al., 2017). Nam et al. (2021) used style-agnostic networks to reduce domain shift by disentangling style encoding from class categories. Lample et al. (2017) originally proposed using adversarial auto-encoder to swap attributes in the images, and later Lotfollahi et al. (2021) demonstrated how a similar idea based on disentangling information from different domains could be applied for out-of-distribution predictions of unseen drug combinations. The method proposed herein combines aspects of these works in order to find a fast and easy way to train networks on simulated data that can be subsequently applied to experimental data.

### A.2. Deep Learning in Optoacoustic Imaging

Several deep-learning-based methods have been used to enhance the MSOT imaging performance. For example, densely-sampled data was recovered from sparse signal acquisitions using supervised learning (Davoudi et al., 2019; Antholzer et al., 2019; Guan et al., 2020). The inverse reconstruction problem has been tackled by learning the optimal regularization in iterative methods (Hauptmann et al., 2018). A semantic segmentation network was applied to hybrid OPUS images for delineating the mouse boundaries in preclinical data (Lafci et al., 2021). MSOT images are also segmented using convolutional neural networks (Gröhl et al., 2021; Chlis et al., 2020). Multi-modal images from MSOT and magnetic resonance imaging (MRI) systems were registered using segmentation and spatial transformer networks (Hu et al., 2021). Spectral unmixing between different wavelengths in MSOT images was performed using deep learning methods (Olefir et al., 2020; Gröhl et al., 2021). Speed of sound values that are used in MSOT image reconstruction was corrected by a learning based method (Jeon and Kim, 2020). Noise caused by electromagnetic interference in MSOT imaging setups could be removed by means of signal domain learning approach (Dehner et al., 2021). Bandwidth enhancement was also proposed in the signal domain using supervised learning methods (Gutta et al., 2017). MSOT image and signal domain data were combined in hybrid networks to reduce limited-view artifacts (Davoudi et al., 2021; Lan et al., 2019), although signal data was only used as complementary information for the image domain learning. The two-step method proposed herein solely operates in the signal domain to solve limited-view-associated problems.

## Appendix B. Image Reconstruction Methods

Two different methods were used in this study to generate images from time domain signals, namely back-projection and elastic-net (Zou and Hastie, 2005). Back-projection is a simple and widely used method based on delaying and summing the US signals according to their time of flight from the pixels on the reconstruction grid (Ozbek et al., 2013). It is applied as follows: i) the signals are first band-pass filtered between 0.1 and 6 MHz, ii) the filtered signals are normalized around zero mean, iii) the pre-processed signals are summed up based on their time of flights. The field of view (FOV) was set to 25.6 mm (256x256 pixels). The elastic-net method was also used for reconstructing the images (Zou and Hastie, 2005). It is based on regularized inversion considering conventional model-based reconstruction (Dean-Ben et al., 2012a). This regularization approach was chosen as it allows removing some artifacts from the non-regularized version of the model-based algorithm and further enables computationally comparing the effect of geometry correction in the signal domain. We used a fixed parameter for the elastic net, $\alpha = 10^{-5}$. This value was empirically established for providing an optimal quality of the reconstructed images with simulated and experimental multisegment array data.

## Appendix C. Benchmark UNet

For comparison purposes, we trained a UNet model on the simulated image domain that learns how to map from MSOT reconstructions obtained with the linear part of the array to those achieved with the full multisegment geometry. For this, an Adam optimizer and MAE loss was used. As opposed to the original UNet (Ronneberger et al., 2015), we use batch normalization (Ioffe and Szegedy, 2015) with zero padded convolutional layers in order to keep resolution constant within a convolutional block. In addition, benchmark UNet has a single residual connection from input to final model output (Jin et al., 2017) such that only residual mapping needs to be learned, and start with 32 convolutional kernels at full resolution instead of 64, going up to 512 instead of 1024 convolutional kernels at the coarsest level, consisting of a total of four max pooling layers. Below, we show that this baseline does not generalize well to experimental data.

## Appendix D. Training

Two different network architectures were trained to reduce limited-view artifacts in the signal domain. The networks were implemented in pyTorch (v1.9) using CUDA (v11) and cuDNN (v8) libraries. Two NVIDIA Titan X GPUs were used in parallel for training. Both networks were trained for 200 epochs using batch size of 16. Adam optimizer with learning rate of 0.001 and weight decay of 10 were used for loss minimization. Style network was trained by leveraging the combination of six different losses as described in methodology section. The loss functions were weighted heuristically to arrange the effects of each loss. Specifically, weights $\ell_{adv\_Latent}$, $\ell_{patch\_sim}$ and $\ell_{patch\_exp}$ were set to 0.001. $\ell_{FM}$ and $\ell_{Cycle}$ were weighted with 0.1. $\ell_{MAE}$ was directly added with weight of 1. All discriminators are trained with gradient clipping penalty proposed in Arjovsky et al. (2017) to mitigate the potentially too strong of discriminator and poor convergence of the adversarial losses.

## Appendix E. Supplementary Results

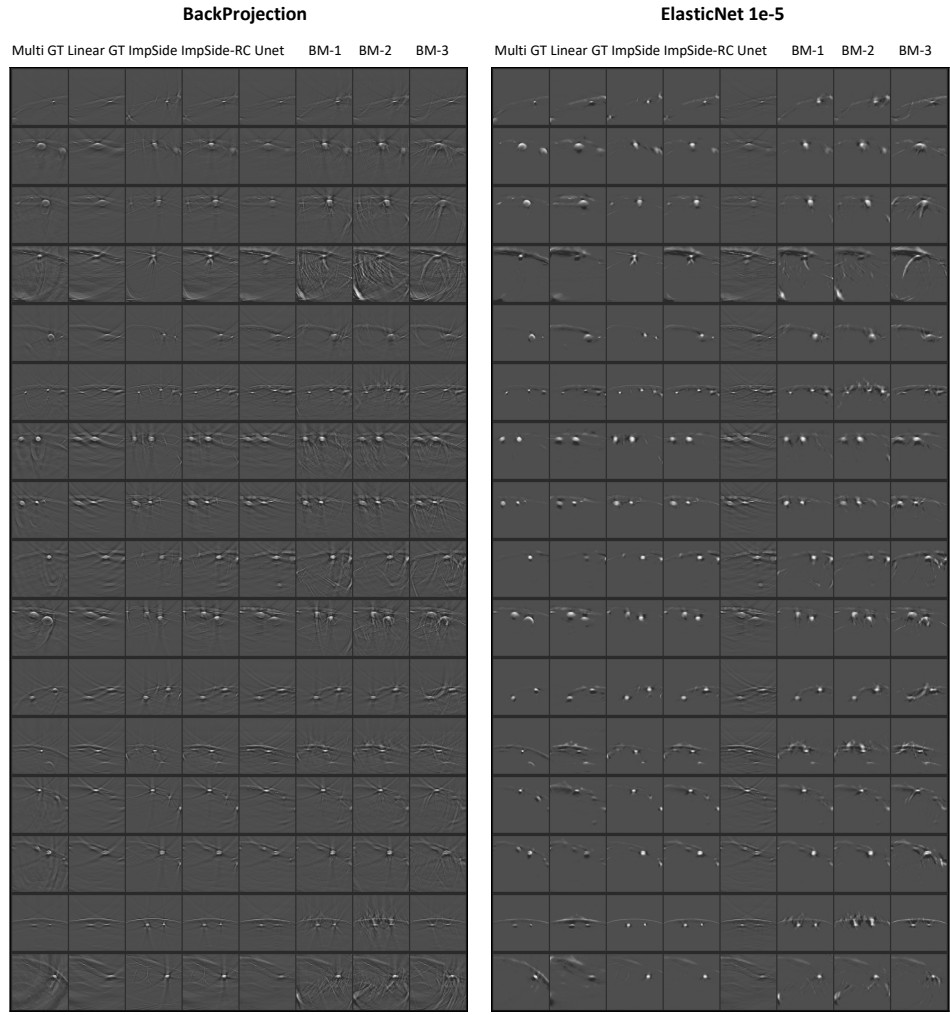

Supplementary Figure 1: Example of reconstructions on the test set (sample number 1-16). Each row corresponds to a different input signal. Each column corresponds to a different method (best viewed digitally). "BM-1" - our sides network applied without prior style transfer network and trained with both synthetic data and linear parts of experimental data, "BM-2" - our sides network applied without prior style transfer network trained only on synthetic data, "BM-3" - supervised sides network which predicts from linear part array signal of concave parts.

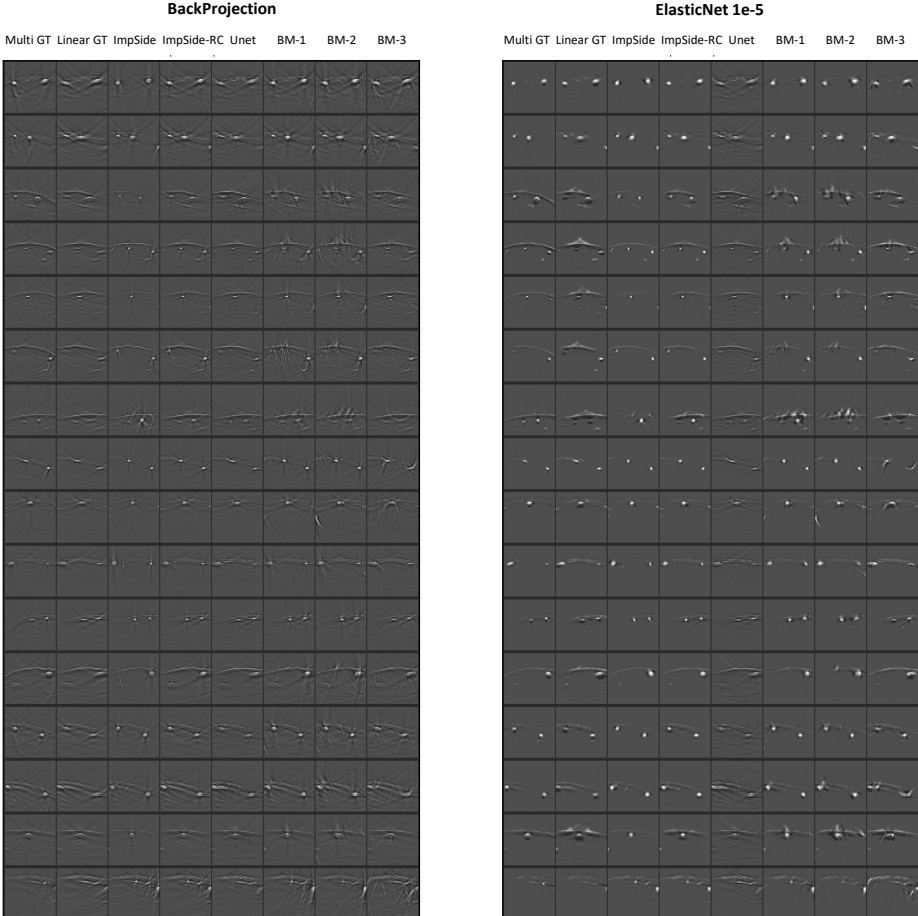

Supplementary Figure 2: Example of reconstructions on the test set (sample number 17-32). Each row corresponds to a different input signal. Each column corresponds to a different method (best viewed digitally). "BM-1" - our sides network applied without prior style transfer network and trained with both synthetic data and linear parts of experimental data, "BM-2" - our sides network applied without prior style transfer network trained only on synthetic data, "BM-3" - supervised sides network which predicts from linear part array signal of concave parts.

Supplementary Table 1: Reconstruction scores with respect to ground truth (GT) multi-segment reconstruction using Elastic Net ($\alpha = 10^{-5}$). "BM-1" - our sides network applied without prior style transfer network and trained with both synthetic data and linear parts of experimental data, "BM-2" - our sides network applied without prior style transfer network trained only on synthetic data, "BM-3" - supervised sides network which predicts from linear part array signal of concave parts. Name of our main proposed method is in bold.

| Dataset/Metrics | SSIM | MSE | Pearson | PSNR |
|---|---|---|---|---|
| Linear GT | 0.85 ± 0.06 | 0.0019 ± 0.0010 | 33.75 ± 2.22 | 75.86 ± 2.22 |
| Side impute | 0.89 ± 0.04 | 0.0018 ± 0.0008 | 33.89 ± 1.93 | 76.00 ± 1.93 |
| **Side impute RC** | 0.90 ± 0.03 | 0.0013 ± 0.0006 | 35.28 ± 2.17 | 77.39 ± 2.17 |
| Unet | 0.57 ± 0.10 | 0.0026 ± 0.0009 | 32.14 ± 1.73 | 74.25 ± 1.73 |
| BM-1 | 0.86 ± 0.05 | 0.0019 ± 0.0011 | 33.92 ± 2.26 | 76.03 ± 2.26 |
| BM-2 | 0.85 ± 0.06 | 0.0025 ± 0.0014 | 32.76 ± 2.40 | 74.87 ± 2.40 |
| BM-3 | 0.84 ± 0.05 | 0.0025 ± 0.0011 | 32.50 ± 2.02 | 74.61 ± 2.02 |

Supplementary Table 2: Reconstruction scores with respect to ground truth (GT) multi-segment reconstruction using BackProjection (BP). "BM-1" - our sides network applied without prior style transfer network and trained with both synthetic data and linear parts of experimental data, "BM-2" - our sides network applied without prior style transfer network trained only on synthetic data, "BM-3" - supervised sides network which predicts from linear part array signal of concave parts. Name of our main proposed method is in bold.

| Dataset/Metrics | SSIM | MSE | Pearson | PSNR |
|---|---|---|---|---|
| Linear GT | 0.67 ± 0.11 | 0.0013 ± 0.0007 | 35.37 ± 2.39 | 77.48 ± 2.39 |
| Side impute | 0.51 ± 0.13 | 0.0025 ± 0.0012 | 32.45 ± 2.02 | 74.56 ± 2.02 |
| **Side impute RC** | 0.64 ± 0.11 | 0.0014 ± 0.0008 | 35.11 ± 2.52 | 77.22 ± 2.52 |
| Unet | 0.49 ± 0.13 | 0.0036 ± 0.0018 | 30.92 ± 2.15 | 73.03 ± 2.15 |
| BM-1 | 0.53 ± 0.13 | 0.0022 ± 0.0017 | 33.43 ± 2.74 | 75.54 ± 2.74 |
| BM-2 | 0.50 ± 0.12 | 0.0025 ± 0.0024 | 32.93 ± 2.67 | 75.04 ± 2.67 |
| BM-3 | 0.54 ± 0.13 | 0.0022 ± 0.0015 | 33.33 ± 2.75 | 75.44 ± 2.75 |

