# OpenReview forum: "Signal Domain Learning Approach for Optoacoustic Image Reconstruction from Limited View Data"
_MIDL.io/2022/Conference — MIDL 2022_

### Official Review · Reviewer_u4UQ · 2022-01-15

**Confidence:** 5
**Preliminary Rating:** 3
**Recommendation:** Poster

**Summary:**

The work introduces a two step deep learning based technique for recovering the MSOT images from tomographic data which can be acquired in sub-optimal geometries. It shows that the sub-optimal geometries can be used for acquiring the data and this can be corrected only in the signal domain using the DL based methods.

**Strengths:**

A deep learning based model was developed for recovering the MSOT images. The method was validated on a number of phantoms for verifying the proposed technique and was found to get better results for sub-optimal geometries.

**Weaknesses:**

The results are applicable only for very simple geometries and it is tough to see the translation to complex phantoms. Also, the comparisons are shown for the very basic model although  a lot of better models are proposed for improving the reconstruction.

**Deanonymize Review:**

no

**Detailed Comments:**

1. The literature review is not strong. I will be interested in looking into how the problem is relevant as well as what other researchers have done in this domain but currently it contains the literature from the same group.

2. The proposed method needs to be compared with state-of-the-art networks. UNet is good for initial comparison but there are many architectures which beat the UNet architecture and a comparison with those will be more helpful in realizing the potential of the proposed method.

3. The test set consists of only 32 images. This is too less for verifying the results of the proposed method. The training is also performed on a very small dataset. Since the details of UNet is not available , I believe it has very high number of parameters and thus it is tough to say that the model is trained properly. It might be under-fitted.

4. The simulated dataset does not have details of the training, testing, validation split.



**Final Rating After The Rebuttal:**

3: Borderline

**Justification Of The Final Rating:**

I am still not satisfied with the revisions from the authors as I feel there should be more comparison with the other methods which will strengthen the work. Also the current test dataset size is not sufficient to justify the results obtained.

In the current revised manuscript, I cannot raise my rating of the work.

**Paper Type:**

methodological development

**Questions To Address In The Rebuttal:**

1. A strong literature review will help in getting the contributions from this paper to be visible. Please outline the contributions explicitly how the work is different from the previous works proposed.

2. Please compare with advanced state-of-the-art methods with the proposed technique to show the clear improvement.

3. Please do an evaluation on a large dataset. The testing dataset is too small. Also the training dataset for UNet is very small for training the model.


**Special Issue:**

no

---

### Official Review · Reviewer_yaRM · 2022-01-21

**Confidence:** 2
**Preliminary Rating:** 1
**Recommendation:** Poster

**Summary:**

The paper describes a method to recover missing tomographic signals from multi-spectral optoacoustic tomography (MSOT) data.  The details of the work may be understood by an expert on MSOT, but are not explained well for the broader MIDL audience.  This made it hard to understand what experiments were being performed or why the results might be interesting.

**Strengths:**

It's good that the paper has access to a large number of experimental images.  The results do seem to show some cosmetic reduction in streak artifacts, although I don't think this cosmetic improvement has any practical significance (it is easy for humans to identify streak artifacts, and there is no difference in the visual interpretation of the BackProjection and ElasticNet images)

**Weaknesses:**

It was difficult for me to identify the core novelty of this work, and I don't think the results were presented very clearly or convincingly.  The authors did report improved numbers, but I don't see a big difference in the practical usefulness of the images, and it was hard for me to understand if the improvement was enabled by an especially creative or novel insight.

The introduction takes too long to describe the contribution of the paper.  There are a lot of unnecessary details about transducer design that don't seem related to the rest of the paper, and there is not enough description of existing reconstruction methods to put the contribution in context.

The paper seems to be combining several existing ideas (adversarial training, feature matching, ...), and it was difficult to identify the key innovation.

The concept of the "multisegment array" and its "right and left concave parts" are not explained clearly enough.  The paper should not assume that readers are familiar with MSOT concepts and jargon and should either explain details like this if they're important or otherwise omit them.  It is not clear why virtual signals need to be imputed at the right and left parts of the multisegment array -- what is the benefit of imputing these signals rather than measuring them directly, and why are they needed at all.

The motivation for this approach is also unclear -- the results suggest that it is possible to measure ground truth, so why even bother with imputation?

There were no experiments demonstrating that style transfer/domain adaptation was important.

The RC and OAT and GT acronyms are used in the results but never defined or explained.

The results show a benchmark UNet without any citation or explanation of what it attempts to do or how it was trained.  It is not clear if this is a fair or interesting comparison.

There are substantial appendix materials that the main paper never refers to.  This organization is unacceptable (very difficult to read).

**Deanonymize Review:**

no

**Final Rating After The Rebuttal:**

4: Weak Accept

**Justification Of The Final Rating:**

I'd like to thank the authors for their careful revisions -- the new version of the manuscript is much easier to understand, and the claims of the manuscript have been sufficiently toned down that I feel comfortable to substantially improve my rating.

**Paper Type:**

methodological development

**Questions To Address In The Rebuttal:**

The paper needs to be rewritten so that it is more accessible to readers who are less familiar with multi-spectral optoacoustic tomography, and also needs to do a substantially better job of identifying the key innovations.

**Special Issue:**

no

---

### Official Review · Reviewer_7n2q · 2022-01-25

**Confidence:** 2
**Preliminary Rating:** 5
**Recommendation:** Oral

**Summary:**

The submitted manuscript titled "Signal Domain Learning Approach for Optoacoustic Image Reconstruction from Limited View Data" presents a deep learning (DL)-based approach to integrate simulated data from multi-spectral optoacoustic tomography (MSOT) for improved reconstruction of MSOT images for two types of probe configurations. The authors approach the scientific questions by two steps, first a network for style transfer from simulated to real, measured MSOT data and second a supervised training on simulated data. The resulting two step solution allows to recover missing tomographic signals on realistic data.

**Strengths:**

MSOT is a promissing imaging method which can, due to the large amount of acquired data benefit from DL-based reconstruction methods. The concept to separate the problem in two sub-problems, namely the problem to close the domain gap between simulations and experimental data and the problem to have well controlled training data to enhance the tomographic image, is well thought through and shows promissing results. The choose networks are modified versions of established network architectures.


**Weaknesses:**

The presenet work is not discussed openly nor are shortcomings mentioned to a realistic level (from my point of view I would expect to read more about the limits and unsolved challenges). The results in all metrics (SSIM, MSE, MSA or PSNR) are not as clear as I would have assumed. Table 1 and Table 2 (appendix) also rise the question how much it depends mainly on the reconstruction (elastic net vs. BackProjection).

**Deanonymize Review:**

no

**Paper Type:**

methodological development

**Questions To Address In The Rebuttal:**

In my opinion the submitted manuscript has some shortcomings and weaknesses, these are mainly in the evaluation and results section and could have been addressed in a broader discussion instead of the rather short and one-sided conculsion statement.
E.g . the differences in the image reconstructions: Results in Fig.3 should have been pointed out and potential reasons for the differences (e.g. scatter apprear more roundish in Side Impute with RC).

**Special Issue:**

yes

---

### Official Review · Reviewer_8GBZ · 2022-01-28

**Confidence:** 3
**Preliminary Rating:** 2
**Recommendation:** Poster

**Summary:**

This paper proposes a method for reconstructing optoacoustic images from limited view data. The method first synthesizes experimental quality raw signals from simulated examples via a style transfer approach. These synthesized examples are used to train a second network, which takes in data from the linear part of a signal array and decodes it into the three parts of a multisegment acquisition; the trained network can then be used on in-vivo linear data to infer the side array data, which can be used to obtain a higher-quality reconstruction. The method is developed using simulated cross-sectional images of an arm-like curved structure and acquired forearm images, and then is tested on a separate held out set of forearm images.


**Strengths:**

- **The proposed method appears to yield good results when reconstructed using ElasticNet.** I appreciate the inclusion of all 32 test set reconstructions; visually, the ImpSide-RC reconstructions generated with ElasticNet do look better to me than the other methods (though I am not an expert on this type of imaging and may be missing details here).
- **The motivation of the paper is clear.** Being able to infer multisegment data from a linear MSOT acquisition is a useful and interesting idea.

**Weaknesses:**

- **An ablation study is needed to justify the complexity of the proposed method.** This paper is missing evidence that the style transfer portion of the method is useful. To demonstrate this, we would need to see that, if parts (B) and (C) in Figure 2 are trained alone without the preceding style transfer step, the results are worse than if style transfer is included. Also, I believe from section 4.1 that several thousand cross sections of multisegment in vivo data are available. If so, to justify the inclusion of the simulated data, another comparison needs to be made to a network like the one in Figure 2(B) trained using only the training set of the in vivo data.
- **Some design decisions are not justified in the text.**
    1. Why does the decoding procedure differ for simulated and experimental data in the style transfer network? I’m particularly referring to this sentence: “If the label is “experimental”, a latent vector is sampled from a Gaussian distribution with mean and variance parametrized by embedding layers in the decoder, and then a convolutional network is applied to it. Why is this necessary in the case of experimental data, but not simulated data?
    2. If multisegment in vivo data are available, why is only the central section used for training the side network?
    3. Why does the center decoder need to be trained in Figure 2(B)? The experimental results are better when the real center is used instead of the decoded center, if I’m understanding the “RC” label correctly — so it is unintuitive that this center decoder is trained at all. This is especially true if multisegment in vivo data are available.
- **The writing is confusing/contradictory in some sections which make it difficult to understand the method, and does not prioritize the key results.**
    1. The main quantitative results are currently in Appendix E and need to be moved up to the body of the paper. The introduction can be made much more concise to make space for this.
    2. Equations 3, 7, and 8 define loss terms which are not used in equation 9 — is this an omission, or are these used in some other way? Either way, this needs to be clarified.
    3.  A few less critical corrections in Detailed Comments, below.

**Deanonymize Review:**

no

**Detailed Comments:**

- In some places, the subscript ‘inv’ is used to represent experimental data, while in other places, ‘exp’ is used — please make this consistent (in particular, Eqns. 9  and 10).
- Please be more specific in specifying $D_{\theta_{disc}}$ in the section above Equation 1; this is confusing later in the text. In Equations (2-3), $D_{\theta_{disc}}$ refers to the discriminator on the latent space, while in Equations (7-8), this same expression appears to be used to refer to the discriminator on the decoder output.

**Final Rating After The Rebuttal:**

5: Strong Accept

**Justification Of The Final Rating:**

The authors have carefully addressed all of the questions and concerns I raised in my initial review, especially the need for ablation studies which demonstrate that each component of the proposed solution is useful, and I do not have outstanding concerns.

I have also seen other reviewers' concerns regarding the small size of the test dataset and comparisons to other methods. I understand these concerns, but have decided not to penalize the paper for them. While experiments on a larger dataset would strengthen the paper, the results provided in the manuscript look consistently better than the comparisons across almost all samples (though I am not an expert in reading this type of imaging data).

Also, the authors have performed a careful ablation study which I believe allows readers to fairly assess the components of the system. In my opinion, this comparison is more informative to a reader about which parts of the system are contributing than a comparison to other state-of-the-art methods, which would be harder to perform in a controlled way. Also, with a quick literature search, I was unable to find previously proposed deep learning solutions to this particular problem which are not essentially a version of the baselines studied in the paper (though again, I have never specifically studied this kind of imaging).

**Paper Type:**

both

**Questions To Address In The Rebuttal:**

- Have I missed a piece of experimental evidence showing that the style networks and/or simulations improve performance?
- Why does the decoding procedure for experimental data involve sampling from a Gaussian, while the decoder for simulated data does not?
- Why is only the central section of the in vivo data used for training the networks?

**Special Issue:**

no

---

### Meta-Review · Area_Chair_VyXf · 2022-02-16

**Recommendation:** Accept (Oral)
**Confidence:** 4

**Metareview:**

The paper received four reviews that at large agree on acceptance. The revision, including editorial changes regarding clarity and accessibility, together with the inclusion of a careful ablation study satisfied most reviewers' concerns.
The remaining concerns are with a limited dataset and choice of baselines, which may be acceptable at this stage.

---

### Decision · Program_Chairs · 2022-02-28

Accept